# Assessing the Cleanliness of Dental Implants Using Scanning Electron Microscopy and Energy-Dispersive X-ray Spectroscopy Analysis—A SEM and EDS In Vitro Study

**DOI:** 10.3390/jfb14030172

**Published:** 2023-03-22

**Authors:** Tarek Mtanis, Ameer Biadsee, Zeev Ormianer

**Affiliations:** Department of Oral Rehabilitation, The Maurice and Gabriela Goldschleger School of Dental Medicine, Tel Aviv University, Ramat Aviv, Tel Aviv 6997801, Israel

**Keywords:** cleanliness, dental implants, osseointegration, surface contamination, titanium

## Abstract

A wide variety of titanium (Ti) alloy dental implant systems are available and as a result, choosing the correct system has become a challenge. Cleanliness of the dental implant surface affects osseointegration but surface cleanliness may be jeopardized during manufacturing. The purpose of this study was to assess the cleanliness of three implant systems. Fifteen implants per system were examined with scanning electron microscopy to identify and count foreign particles. Particle chemical composition analysis was performed with energy-dispersive X-ray spectroscopy. Particles were categorized according to size and location. Particles on the outer and inner threads were quantitatively compared. A second scan was performed after exposing the implants to room air for 10 min. Carbon, among other elements, was found on the surface of all implant groups. Zimmer Biomet dental implants had higher particle numbers than other brands. Cortex and Keystone dental implants showed similar distribution patterns. The outer surface had higher particle numbers. Cortex dental implants were the cleanest. The change in particle numbers after exposure was not significant (*p* > 0.05). Conclusion: Most of the implants studied were contaminated. Particle distribution patterns vary with the manufacturer. The wider and outer areas of the implant have a higher probability of contamination.

## 1. Introduction

Installing dental implants is an integral part of the daily practice of most dental clinics [1]. Due to the large number of implant systems offered, it is challenging for dentists to choose the best system for their patients.

The success of dental implants is largely attributed to osseointegration (OI), a term originally defined by Branemark in the 1950s, that implies direct histological contact between living bone and the implant, as seen in light microscopy [2,3]. Zarb & Albrektsson described OI as the “process whereby clinically asymptomatic rigid fixation of alloplastic materials is achieved and maintained in bone during functional loading” [4]. 

Ti and its alloys are the materials of choice for dental implants due to their low specific weight and elastic modulus, and high corrosion resistance and biocompatibility [5]. OI of Ti is related to its passive, dense, and resistant oxide layer, which protects the underlying metal from corrosion and, in turn, allows the desired biological response of new bone formation [5]. This process begins with the absorption of ions, proteins, polysaccharides, and proteoglycans by the Ti oxide layer. Subsequently, macrophages, neutrophils, and osteoprogenitor cells migrate to the bone-implant interface and lead to bone apposition in close contact with the implant surface [6]. With that in mind, implant systems have evolved to achieve better and faster OI. Over time, the adoption of surface modification techniques for implants has led to better clinical outcomes [7]. Surface modifications can be additive, such as Ti-plasma spraying ion deposition, or subtractive, such as abrasive blasting and etching [8].

Two main parameters contribute to the stability and reactivity of implants to achieve successful OI [2]. First, the macroscopic design of the implant allows for the primary stability required for the biological process of OI [9]. A lack of primary stability is a risk factor for early implant failure [10,11,12]. The second parameter is the implant’s surface properties, such as chemical composition, topography, charge, and degree of wettability [2,13]. Because even small amounts of surface contamination can impair OI, it is important that the surface has the lowest possible contamination count, especially if it is preventable [14]. Nevertheless, relatively little is known about the effects of specific impurities at low concentrations. 

Hydrophilicity is directly related to osseointegration, especially in the initial phase; a hydrophilic implant surface attracts more water and ions, resulting in stronger bonds between the implant surface, proteins, and cells; the more stable the adhesions, the better the osseointegration [15]. A lack of cleanliness and the presence of surface impurities decrease surface energy and hydrophilicity; consequently, less biomolecular contact leads to less bone-implant contact, which impairs healing and OI [16,17,18,19]. In addition, surface impurities worsen the immunological foreign body response, which produces free radicals. Consequently, corrosion and ion release are accelerated. 

Cleanliness is defined as when the number of impurities is below a certain level, and cleaning is the process of removing biological, chemical, and physical contaminants [20,21]. The American Society of Testing and Materials has published a protocol for cleaning surgical implants without solvents [22]; starting with using alkaline systems to remove oil and lubricants, then followed by sandblasting with high-quality, iron-free silica. This step is usually followed by acid pickling if the sandblast material contains iron. The next step in the protocol is removing carbon, nitrogen, and oxygen. It is recommended to perform controlled acid pickling with different concentrations of sulfuric, nitric, or hydrofluoric acid. However, implant manufacturers may use variations of this protocol [23]. Keystone dental implants have been reported to have a “clean enough” surface due to the use of ultra-pure water as the final step in the cleaning process [24].

Marginal bone loss and the clinical performance of dental implants depend on factors that usually cannot be monitored; direct evidence of damage caused by impurities cannot be determined. However, impurities are thought to affect bone formation and OI, alter implant survival, and cause toxicity and systemic allergic reactions [5,25,26,27,28,29,30]. Peri-implantitis (PI), for example, is a major problem in the dental field; progressive bone loss can eventually lead to the loss of the implant. Corrosion plays an important role in the etiology of PI [17]. It leads to the release of Ti ions into the surrounding tissue and induces an immune response, which in turn can lead to bone resorption [31,32].

At all stages from fabrication to placement of the implant into the bone, the likelihood that the implant surface will become contaminated with metals, lubricants, or other chemical compounds is high; this can be caused by milling burs/machines, surface modification materials, sterilization, packaging/storage, dentist handling, or even aging [5,33]. To address this matter, the International Organization for Standardization (ISO 19227: 2018) prescribed acceptance criteria for carbon contamination on orthopedic implants; the total carbon acceptable on medical devices is less than 500 µg [34]. However, detailed requirements for evaluating the cleanliness of dental implants are not available. Despite being approved for human use, several dental implants had unacceptable levels of contamination—through Energy dispersive X-ray spectroscopy (EDX), organic and inorganic residues were found on newly unpacked implants; the contaminants were thought to originate from the manufacturing, handling, and packaging [26]. To address this issue, the Clean Implant Foundation published a consensus paper describing objective criteria for evaluating surfaces. According to these criteria, single or numerous organic particles are typically found, and systematically distributed on implant surfaces. The recommended threshold is up to 5 particles smaller than 50 µm on each scan, or up to 15 particles on the entire surface. Elements such as iron, nickel, chromium, copper, tin, tungsten, antimony, fluorine, or carbon are not acceptable. However, blasting material residues of Ti or aluminum oxides are [35].

EDX is a standard method frequently used in dental research to analyze the elemental compositions in organic and inorganic material samples [5,14,25,30,36,37,38,39]. An electron beam projects toward the sample‘s surface, excites it, and causes the emission of variable X-ray photons; an energy-dispersive X-ray spectroscopy detector (EDS) identifies the distinctive energy of each element [40]. 

The purpose of the present study was to detect, count, and identify contamination of implant surfaces using SEM and EDS. The surface of implants also attracts organic contaminants from free air. Therefore, they are susceptible to contamination during the period between unpacking and implantation [28]. A second scan was conducted to verify whether the number of organic contaminants changed significantly after the implants were exposed to room air for ten minutes.

We hypothesized that dental implants approved for human use may be contaminated. We also hypothesized that there is no significant change in the organic particle count on an implant’s surface after exposure to air for ten minutes.

## 2. Materials and Methods

### 2.1. Materials

One implant system from three different implant manufacturers was selected for the study: Dynamix implants (Cortex, Dental Implants Industries, Ltd., Shlomi, Israel), Advanced+ implants (Keystone Dental, Inc., Indianapolis, IN, USA), and Osseotite Tapered Certain implants (Zimmer Biomet Holdings, Inc., Parsippany, NJ, USA). A total of 15 implants per company were examined using a JSM-IT100 SEM (JEOL USA, Peabody, MA, USA). Five implants were ordered from the factory and ten additional implants were purchased from random practices to ensure that samples were randomly selected.

### 2.2. Methods

SEM imaging and EDX analysis were performed on three predefined areas of each implant. The shoulder (Sh) is the coronal part of the implant that usually has the largest diameter. The body (B) is the middle part and has the largest surface area. The apex (A) is defined as the apical 3–4 mm of the implant. Each part is divided into an inner and an outer area, namely, the root and the thread crest, respectively (Figure 1).

### 2.3. Unpacking the Implants

The operator wore a sterile mask and gloves. Without touching the surface, each implant was removed from its packaging with a sterile insertion instrument and fixed on the specimen holders with carbon tabs. During this procedure, the implant surface to be examined had no contact with other materials. The estimated time from opening the implant packaging to closing the SEM chamber with the implant inside was 10 to 30 s.

### 2.4. Image Mapping

Implants were scanned at 30× to 400× magnification from the shoulder to the apex. Backscattered electron mode (BSE) was used to quantitively measure suspected contamination in different areas of the implant surface. BSE mode provided compositional information during the imaging phase. Low atomic number elements such as carbon and silica appear dark, while high atomic number elements such as Ti appear brighter.

### 2.5. Counting and Detection of Impurities

Foreign materials and suspected impurities were counted and analyzed. Starting with a magnification of 30×, all suspected impurities were marked and counted. The smallest particle that can be clearly identified at 30× is 20 µm in “diameter”. Since the geometric shape of the particles was not constant and varied between irregular shapes resembling circles, rectangles, and squares, the “diameter” was calculated as the mean between two-thirds of the length and one-third of the width. Then, near one of the impurities found at 30×, the smaller and not clearly obvious particles were inspected again with 100× magnification and identified as medium (M) sized. In addition, 200× magnification was used to examine a randomly selected area on the outer thread of the implant and then on the inner implant surface to detect differences in the distribution of particles in two adjacent areas with different surface depths.

The contaminants were classified into three size-dependent groups: Large (L), Medium (M), and Small (S). Group L included impurities of ≥20 µm diameter that were identified with 30× magnification. Group M included impurities with a range of 8–20 µm that were identified with 100× magnification, and group S included impurities <8 µm that appeared as small circular dots with 100× magnification. A 200× magnification was used to compare implant inner and outer surfaces while combining large and medium particles into one group (LM).

### 2.6. EDS Scanning Parameters

As recommended in the literature for scanning surfaces and detecting organic elements, low beam energy (5 kV) was used to detect the organic components of the surface contaminants. The working distance was 10 to 14 mm. For better accuracy, a dead time (DT) of 20–30% was targeted. As a result, in some cases, the probe current was adjusted when the DT was too high or too low. An increase in probe current resulted in an increase in DT and vice versa. The default setting for probe current was 30 s acquisition time and 400× magnification. Spot analysis was performed twice: once to scan the contaminants and once to scan adjacent clean areas. A ZAF correction model was applied for quantification. Results were standardless and the percentage of mass content (wt%) was used for statistical analysis [39]. A second scan was performed to find additional large organic contaminations after exposing the implants to room air for ten minutes. 

### 2.7. Statistical Analysis

Data were analyzed using SPSS version 25.0 (SPSS, Inc., Chicago, IL, USA). For comparison among and within the different companies, ANOVA was performed with repeated measures within-subjects factor Shoulder_Body_Apex (Sh_B_A) and between-subjects factor: Company. After finding the square root of the row data in order to achieve a normal distribution. Paired t-tests were performed to compare the particle counts of the inner and outer surfaces of LM and S particles at 200× magnification, within each company. Statistical significance level was set at 5%.

After finding the root square of the raw data to achieve normal distribution, two-tailed t-tests of two samples, assuming equal variance, were performed to check for significant differences in the distribution of large particles after exposing the implants to room air for 10 min. Data were analyzed using SPSS version 27.0 (IBM, Chicago, IL, USA). Statistical significance level was set at 5%.

## 3. Results

### 3.1. Cortex, Dental Implants Industries, LTD

At 30× magnification, a mean of 4.38 L particles were found per implant. At 100× magnification, there was a mean of 2.92 M particles and 5.68 S particles. At 200× magnification, a mean of 2 LM and 7.73 S particles were found on the outer surface of the implants, while the inner surface mean was 0.36 LM and 5.35 S particles. Some particles had a uniform, well-defined, black surface, while others had a heterogeneous appearance with small and lighter particles (Figure 2).

At 30× magnification, L particles were significantly fewer on the apex compared to the body and shoulder (*p* = 0.03). The M and S groups did not show any significant differences between the areas at 100× magnification. The shoulder had the most particles, in both groups (Table 1).

At 200× magnification, on the outer surface, the shoulder and body had the highest LM number of particles (*p* = 0.034, Table 1). The shoulder had the most S particles on the internal surface (*p* = 0.014, Table 1). The shoulder and body had more particles in most groups. The difference between the external and internal surfaces was significant in all areas, except for the S particles on the shoulder and the LM particles on the apex (*p* < 0.05, Table 2).

Analysis of the particle elemental weight concentration (wt%) showed that four elements dominated on most implants: carbon, nitrogen, aluminum, and oxygen. Carbon was detected on 13 implants at a mean of 57.3% per particle. Particles on 13 implants had a mean of 44.51% nitrogen per particle. Particles detected on 14 implants had 19.47% aluminum per particle. Fourteen implants had a mean of 27.27% oxygen per particle. Five implants had impurities with 87.21% titanium per particle. The other elements were detected in four or fewer implants with less than 10% per particle (Table 3). 

### 3.2. Keystone Dental Inc, USA 

At 30× magnification, a mean of 5.4 L particles were found per implant. Magnification of 100× revealed a mean of 3.35 M particles and 10 S particles. At 200× magnification, a mean of 2.71 LM and 11.45 S particles were found on the outer surface. On the inner surface, a mean of 0.2 LM and 2.51 small particles were found. As with the Cortex dental implants, the apex consistently had a low particle count, but the particle count was higher for the Keystone dental implants compared to the Cortex dental implants. No pattern in the particle distribution was found. Some particles were found as the sole contaminant, while others were surrounded by smaller particles (Figure 3).

The shoulder and body had the highest particle counts at 30× and 100× magnification. The difference between the three areas of the implant was not significant in any group (*p* > 0.05, Table 4). The difference between the external and internal surfaces was significant in all areas (*p* < 0.05, Table 2). 

Analysis of the particle elemental weight concentration (wt%) showed that four elements—carbon, nitrogen, aluminum, and oxygen—dominated on most implants. Carbon was detected on all implants with a mean value of 42.9% per particle. Thirteen implants had a mean of 40.3% nitrogen per particle and 13 implants had particles with a mean of 24.2% aluminum per particle. All implants had a mean of 27.7% oxygen per particle. The other elements were detected in five or fewer implants with less than 15% per particle (Table 5).

### 3.3. Zimmer Biomet Holdings

At 30× magnification, a mean of 10.72 L particles was found. At 100× magnification, a mean of 4.56 M and 16.77 S particles were found. At 200× magnification, a mean of 4.66 LM and 21.65 S particles were found on the outer surface. On the inner surface, a mean of 0.81 LM and 8.85 S particles were found. One implant had a large organic particle with a diameter of 285 µm (Figure 4). Several implants showed a systematic distribution that appeared as if they had been touched on the outer surfaces, like the threads and the apex (Figure 5). The shoulder area had significantly fewer L particles than the body and apex (*p* < 0.01, Table 6). At 100× magnification, there was a significant difference between the areas, due to the higher S particle number on the shoulder (*p* = 0.043, Table 6). At 200× magnification, on the internal S group, the shoulder and apex areas had significantly higher particle numbers than the body (*p*= 0.002, Table 6). The difference between the external and internal surfaces was significant in all areas (*p* < 0.05, Table 2).

Analysis of the particle elemental weight concentration (wt%) showed that carbon, nitrogen, and oxygen dominated on most implants. Carbon was detected on all implants with a mean value of 37.5% per particle. Fifteen implants had a mean of 49% nitrogen per particle and 14 implants had a mean of 21.2% oxygen per particle. Particles on three implants had a mean of 35.6% aluminum per particle and another two implants had a mean of 86.1% titanium per particle. The other elements were detected in fewer than 10 implants with less than 15% per particle (Table 7).

### 3.4. Differences in Particle Counts among Companies

There were significantly more L and S contaminants on the Zimmer Biomet dental implants than on the other two brands (*p* < 0.01, Table 8). The difference in the number of M particles between implant groups was close to significant (*p* = 0.052, Table 8). however, the difference between Cortex and Zimmer Biomet dental implants is large while the difference between Keystone dental implants and the two brands is considered neutral. Cortex dental implants had the cleanest surface.

The particle distributions found at the different magnifications in each area were compared between implant companies. The Zimmer Biomet dental implants had significantly higher particle numbers in most areas (*p* < 0.01, Table 9).

M and S contaminants showed a consistent distribution pattern in all companies; the highest and lowest counts within each company were in the same area. Cortex and Keystone dental implants showed similar distribution patterns and differed from Zimmer Biomet dental implants; L and M contaminants in the apex were lowest in the Cortex and Keystone dental implants. However, M contamination at the shoulder was highest in all systems and L contamination at the body was high in all systems (Table 9). 

### 3.5. Differences in Particle Counts after Implants Were Exposed to Room Air

The scan for large particles, performed after the implants were exposed to room air, showed no significant difference from the first scan in all implant groups (*p* > 0.05, Table 10).

## 4. Discussion 

The objective of this study was to detect and count organic impurities, especially carbon, on implants. The EDX results showed that almost every implant had carbon residue on the surface, which is consistent with most studies [19]. In addition to the expected elements such as Ti, aluminum, and oxygen, other foreign elements were detected in small amounts on a few implants. 

In a study examining implants from Zimmer Biomet Holdings, a carbon content of 3.9% was found on the surface, while the current study found 37.5% [17]. This difference can be attributed to different analysis parameters, such as acceleration voltage and sampling method. In this study, 5 kV and spot sampling with 400× magnification were used, while Rizo-Gorrita et al. [41] used 20 kV and areal sampling with 3000× magnification. In addition, analysis of suspected carbon contaminants identified using BSE mode is more accurate than random area sampling. Similar to the current study, Massaro et al. [5] found traces of various elements such as aluminum, iron, sodium, silicone, etc., with EDS, but the percentages were not reported. However, X-ray photoelectron spectroscopy (XPS) was also used. It revealed 53.7% carbon. This is logical because XPS analyses the top surface of the sample. With EDS, a lower accelerating voltage results in a shallower sample depth and thus, similar results. In contrast to this study, EDX and XPS analyses demonstrated a “clean enough” surface for the Keystone dental implants, which is attributed to cleaning the implants with ultra-pure water [24]. In the current study, most implants had foreign materials on the surface. Two Keystone implants were considered clean, with a maximum of two large particles, this difference may be attributed to the much smaller sample size in the previous study [24]. 

It is suggested that dental implants inevitably contain some type of foreign material on the surface [24,26,42,43,44,45,46]. Harloff et al. [27] examined the raw material of implants and consistently found small traces of foreign materials. Therefore, the foreign bodies found on implants in the current and other studies may be attributed in part to the raw material. 

The main difference between this study and others is the number of implants studied. Fifteen implants per system were examined, whereas in other studies the number of implants ranged from one to eight. In addition, each implant was examined independently in three different areas with repeated measurements and increasing magnification. In one study, six implants per system were examined in three different areas, as in this study; however, the total area of interest was smaller (4.5 mm^2^) than the surface area scanned in this study [44]. Other studies examined one implant per system, but they examined the total surface area scanned [5,45]. The number of implants per system in the present study was chosen according to the recommendation of statisticians. Implant companies recommend at least five to seven implants to make a valid statement about the quality standard [24]. 

Contaminants of various sizes have been traced for diagnostic purposes; large and medium contaminants are believed to be the result of sputtering, air contamination, and point contact [5]. “Plaque-like” contaminants, which are small and scattered, are believed to be the result of packaging leaks or a larger area of contact, such as when the implant surface touches or is near the packaging walls when implants are pulled against the walls of the packaging (which is unavoidable). Due to the contaminants’ geometric heterogeneity in this study, the diameter was calculated using their length and width mean, whereas other studies did not go into detail about this issue [24,26,42,44]. in the current study, most large organic contaminants had a maximum diameter of 50 µm. However, one of the Zimmer Biomet implants had a single 285 µm diameter particle. Duddeck et al. [26] detected organic contaminants with a similar diameter range. However, the largest particle was 70 µm diameter. Duddeck et al. [42] also examined zirconia implants. Organic particle diameters ranged from 5 to 60 µm and the number ranged from zero to more than 50. One company showed a clean surface in all three samples. Schupbach et al. [44] used surface area to evaluate particle size. The mean size was 1120 ± 1011 µm^2^ and the largest particle was 5900 µm^2^. The mean particle number reported was 17.7. However, the results are mainly related to aluminum. 

In this study, different areas of each implant were examined separately (Sh, B, A) to check whether the amount of contamination was area dependent. Guler et al. [25] also examined different areas separately, but used different parameters and up to 3000× magnification. The current study found that contaminants were distributed differently in different areas and implant groups. Moreover, the number of contaminations was size-dependent; smaller contaminations were found more frequently than larger ones in all companies (Table 1 , Table 4 and Table 6). The results cautiously suggest that all contaminations affect all areas. However, this varies with the manufacturer.

If leakage is the reason for small contaminants, then it is more likely to affect the shoulder and body. According to this hypothesis, leakage could be a major cause of small impurities on the Cortex and Keystone dental implants. In addition, it is likely that contact with the interior of the package leads to contamination of the other areas and especially the external threads. The results of this study also suggest that contact with the packages wall is a reasonable cause of contamination of the Cortex and Keystone dental implants (Table 2 and Table 9). Accordingly, Zimmer Biomet dental implants are more likely to be contaminated before or during packaging. Nevertheless, the possibility of a small amount of contamination before packaging cannot be ruled out. 

In the initial stages of healing and OI, the outer surface comes into contact mainly with old bone, while the intraosseous inner surface comes into contact with debris and blood, from which woven bone eventually develops beginning in the first week post-implantation [47,48]. Inner surface contaminants that impair hydrophilicity will affect the first stages of OI [6,15,47]. Accordingly, it may be speculated that inner surface contaminations may affect OI to a larger degree. In quantitative terms, the greater the contamination, the more damage it will cause in all areas. In this study, the outer surface had significantly more contaminants in most areas, which could be a positive finding if this assumption is true.

Various compositional analysis methods are available on the market. In this study, SEM with a built-in EDS was used due to the simplicity and reliability of this combination as a multimethod for detecting, assessing, and compositional analysis, either to suspected contamination on implants as in the current study or to other devices and materials used in the dental field [19,36,38]. The EDX analysis revealed that almost every implant has carbon residuals on the surface, findings that agree with most studies [19]. In this study, the mass percentage calculation was used, as recommended in the literature. The atomic percentage calculation is less accurate because it depends on the assumed atomic weights [25,27]. However, the differences are minor.

Oxygen was found on most implants. It ranged from 3.2% to 66% per particle, consistent with the results of other studies [45]. It is suggested that the origin of this element could be the Ti oxide layer, adsorbed water, and organic compounds [5]. In this study, a quantitative correlation was not found between carbon and oxygen; therefore, the source of oxygen can be attributed to different sources, as mentioned above. For this purpose, the use of XPS is more reliable.

Nitrogen has been found abundantly in suspected contaminants. It has been suggested that its origin can be attributed to Ti nitride, ammonium, and organic nitrogen-containing species [5]. There was substantially more nitrogen in the control spot analysis group of this study. This suggests that the nitrogen found in suspected contamination is less likely to be the result of organic contamination [5].

Although Ti is highly compatible, loose particles are not acceptable due to phagocytosis and migration to other organs [30]. Ti residues are believed to originate from the machining process during manufacturing or from the packaging of the Ti sleeve. Aluminum on most of the Cortex and Keystone dental implants is plausible due to sandblasting with aluminum oxide, while Zimmer Biomet dental implants are not subjected to this process. The use of glass chambers and autoclaving is thought to be responsible for silicone contamination [49]. The silica used for the cleaning process may also be another source of silicon residue. Sulfur and magnesium integration on the surface of the implants results in higher removal torque and increased bone regeneration [25]. Therefore, it is assumed that traces of them are not harmful. Sodium and chlorine found at low percentages may be related to the cleaning of the implant surface with sodium chloride solution [45]. Traces of iron were found on one each of the Cortex and Zimmer Biomet dental implants. It is suspected that iron contaminants are residues from the turning machine or contaminated silica powder used for cleaning, although if this is the case, we would expect to find iron on more implants. 

Using the Clean Implant Foundation criteria to evaluate cleanliness in this study would not be accurate due to methodological differences, sampling method, EDS parameters, and particle diameter measurement. There was no significant difference in large contaminant counts after 10 min of exposure. As a result, one may think that a particle-free, clean room environment is not obligatory for similar studies [26]. To accurately address this issue, it‘s advised that further studies monitor the change in smaller contaminants.

Limitations of this study include the use of EDS only to detect the concentrations of elements. Even though EDX analysis is simple and accurate for detecting elements, it does not determine the origin of the contaminants. In this study, the origin of most elements was estimated in accordance with previous studies. Using additional tools, such as XPS could aid in this matter. For example, XPS can differentiate between atmospheric carbon and carbon from organic contaminants [46]. In addition, the protocol used in this study was developed to facilitate the detection of organic residuals, while considering the limitations of the available tools. Using the “Image-Mapping” mode was not an option due to SEM limitations. This mode allows at least 500× high-resolution images for the entire specimen, making detection and counting easier and more reliable [26]. Measuring the carbon weight with EDS according to the recommendations of ISO is not possible [34]. The use of a 5 kV accelerating voltage in this study is more sensitive to organic materials, leading to higher counts relative to other studies. This may lead to misinterpretation of the results of the current study. The importance of the cleanliness of dental implants is gaining more recognition among practitioners and scientists. Reliable protocols that can be used systematically need to be developed.

## 5. Conclusions

Within the limitations of this study, it can be concluded that the studied implants are contaminated to some extent. Carbon, oxygen, nitrogen, and aluminum are the most common elements. Zimmer Biomet dental implants are the most contaminated. Particle distribution patterns vary with the manufacturer. In general, the shoulder and body have the highest contamination counts. In addition, the outer thread is more prone to contamination. Exposing the implant to room air for up to ten minutes during surgery is acceptable but is not recommended. 

## Figures and Tables

**Figure 1 jfb-14-00172-f001:**
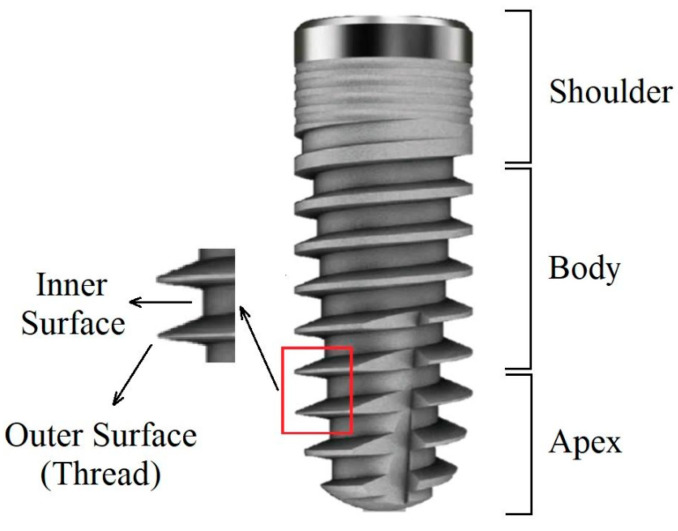
Schematic illustration of the different areas on an implant’s surface.

**Figure 2 jfb-14-00172-f002:**
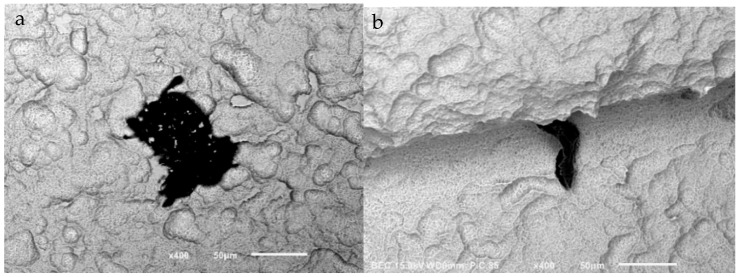
SEM images of two particles at (400×) magnification on Cortex dental implants. Large heterogeneous particle with smaller particles on its surface (**a**) and a uniform, well-defined particle (**b**).

**Figure 3 jfb-14-00172-f003:**
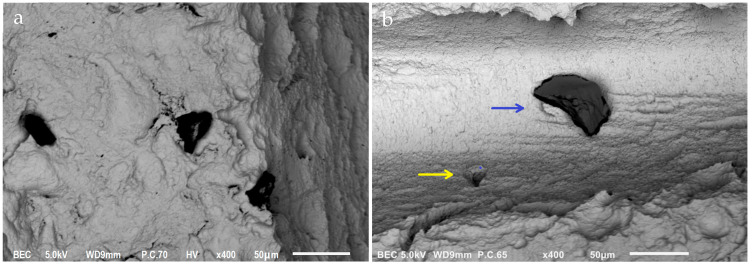
SEM images of two particles at (400×) magnification on Keystone dental implants. Three particles surrounded by smaller particles (**a**) and one large particle (blue arrow) with a smaller particle (yellow arrow) (**b**).

**Figure 4 jfb-14-00172-f004:**
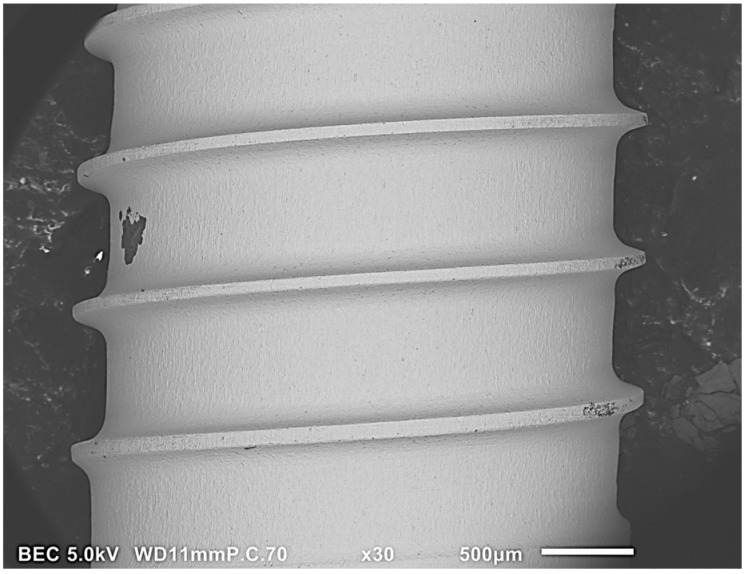
SEM Image (30×) on Zimmer Biomet dental implants. A 285 µm particle with several small particles next to it.

**Figure 5 jfb-14-00172-f005:**
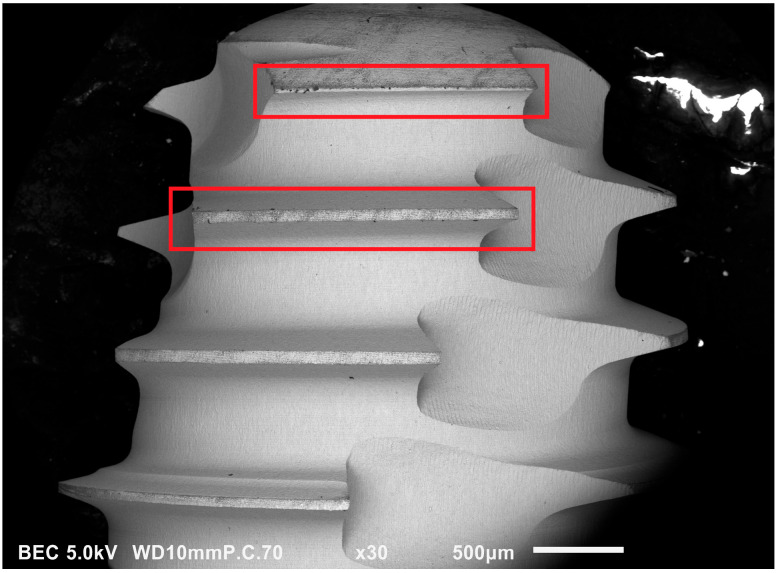
SEM Image (30×) on Zimmer Biomet dental implants. “touch-like” contamination at the apex (upper rectangle) and external thread (lower rectangle).

**Table 1 jfb-14-00172-t001:** Particle distribution on Cortex dental implants. The apex had the lowest particle numbers within most particle size groups. The particle distribution pattern was similar at all magnifications.

Cortex Dental Implant														
Magnification	30×		100×				200× (Outer Surface)				200× (Inner Surface)			
Particle Size Area	L ^1^		M ^2^		S ^3^		LM ^4^		S		LM		S	
	Mean	SD	Mean	SD	Mean	SD	Mean	SD	Mean	SD	Mean	SD	Mean	SD
Shoulder	1.64	±1.05	1.17	±0.91	2.40	±2.28	0.96	±1.17	3.32	±2.85	0.07	±0.25	2.92	±3.47
Body	1.66	±1.35	1.01	±0.69	2.32	±2.17	0.88	±0.9	2.50	±2.18	0.23	±0.47	1.46	±2.03
Apex	1.08	±0.94	0.74	±0.78	0.97	±1.64	0.16	±0.43	1.90	±2.5	0.07	±2.5	0.97	±2.4
Implant mean	4.38	±0.99	2.92	±0.64	5.68	±2.43	2	±1.32	7.73	±2.13	0.36	±0.28	5.35	±3.04
*p*-value	0.03		0.37		0.14		0.034		0.1		0.29		0.014	

^1^ Large particles > 20 µm; ^2^ Medium particles. Between 8 and 20 µm; ^3^ Small particles < 8 µm; ^4^ Large and medium particles > 8 µm.

**Table 2 jfb-14-00172-t002:** The difference between inner (in) and outer (out) surfaces with 200× magnification. The difference in Cortex Implants between the areas was not significant in two groups, the S particles at the shoulder and LM particles at the apex.

Implants Company	Area	S ^1^ (In ^3^–Out ^4^)				LM ^2^ (In–Out)			
		Paired Differences		*t*-Test	*p*-Value	Paired Differences		*t*-Test	*p*-Value
		MEAN	SD±			MEAN	SD±		
Cortex	SH ^5^	−0.40	2.13	−0.72	0.48	−0.90	1.26	−2.76	<0.05
	B ^6^	−1.05	1.62	−2.49	<0.05	−0.65	1.16	−2.17	<0.05
	A ^7^	−0.93	1.53	−2.37	<0.05	−0.09	0.53	−0.69	0.50
Keystone	SH	−2.63	2.76	−3.69	<0.01	−0.67	1.08	−2.97	<0.05
	B	−3.11	2.22	−5.43	<0.01	−1.03	1.34	−3.71	0.01
	A	−3.20	2.73	−4.54	<0.01	−0.82	0.86	−6.30	<0.01
Zimmer	SH	−3.27	2.01	−6.30	<0.01	−1.09	1.49	−2.84	0.013
	B	−5.80	2.04	−11.01	<0.01	−0.79	1.32	−2.24	0.043
	A	−3.73	3.43	−4.20	<0.01	−1.79	1.41	−4.90	<0.01

^1^ Small particles < 8 µm. ^2^ Large and medium particles > 8 µm. ^3^ Inner surface of the implants. ^4^ Outer surface of the implants. ^5^ Shoulder of the implant. ^6^ Body of the implant. ^7^ Apex of the implant.

**Table 3 jfb-14-00172-t003:** Elements distribution and their concentration by mass (wt%) in the particles found on Cortex dental implants with EDS. Silica (Si) was detected on three different implants. Sulfur (S) was detected on one implant in the body. Sodium (Na) was detected on two different implants in the body. Magnesium (Mg) was detected in one implant at the apex. Iron (Fe) was detected on one implant at the apex. Phosphate (P) was detected on four different implants. Vanadium (V) was detected on one implant at the body.

Cortex			
Elements	Mean%	SD	(n) ^1^
C	57.3	±31.62	13
N	44.51	±25.75	13
Al	19.47	±14.3	14
O	27.27	±15.76	14
Ti	87.21	±3.91	5
Si	6.33	± 9.3	3
S	0.76	-	1
Ca	0.00	-	0
Na	0.95	±0.6	2
Mg	2.84	-	1
Cu	0.00	-	0
F	0.00	-	0
Cl	0.00	-	0
Fe	20.67	-	1
P	1.2	±0.45	4
V	66.61	-	1

^1^ The number of implants contaminated with each element.

**Table 4 jfb-14-00172-t004:** Particle distribution on Keystone dental implants. The particles distribution pattern at 200× was different than the pattern at 100× and 30×; the apex had higher counts than the shoulder in most particle sizes (200×).

Keystone Dental Implant														
Magnification	30×		100×				200× (Outer Surface)				200× (Inner Surface)			
Particle Size Area	L ^1^		M ^2^		S ^3^		LM ^4^		S		LM		S	
	Mean	SD	Mean	SD	Mean	SD	Mean	SD	Mean	SD	Mean	SD	Mean	SD
Shoulder	1.63	±1.16	1.38	±1.08	3.72	±2.32	0.80	±0.81	3.30	±2.09	0.13	±0.51	0.67	±1.34
Body	2.24	±1.76	1.18	±1.1	3.53	±2.17	1.03	±1.33	4.19	±2	0.00	-	1.08	±1.68
Apex	1.53	±0.88	0.79	±0.83	2.75	±2.6	0.89	±0.93	3.97	±2.81	0.07	±0.25	0.77	±1.13
Implant mean	5.40	±1.14	3.35	±0.9	10	±1.56	2.72	±0.33	11.45	±1.38	0.2	±0.2	2.51	±0.63
*p*-value	0.06		0.18		0.35		0.83		0.36		0.57		0.35	

^1^ Large particles > 20 µm; ^2^ Medium particles. Between 8 and 20 µm; ^3^ Small particles < 8µm; ^4^ Large and medium particles > 8 µm.

**Table 5 jfb-14-00172-t005:** Elements distribution and their concentration by mass (wt%) in the particles found on Keystone dental implants with EDS. Titanium (Ti) was detected in one implant. Silica (Si) and sulfur (S) were detected in five implants. Calcium (Ca) was detected on one implant at two different sites. Sodium (Na) was detected on three different implants. Magnesium (Mg) and Copper (Cu) were detected on two different implants. Fluorine (F) was detected on one implant on the shoulder. Chlorine (Cl) was detected on one implant on the body.

Keystone			
Elements	Mean%	SD	(n) ^1^
C	42.94	±19.71	15
N	40.32	±13.03	13
Al	24.23	±14.77	13
O	27.67	±12.53	15
Ti	9.09	-	1
Si	5.07	±4.5	5
S	0.89	±0.39	5
Ca	14.6	-	1
Na	1.33	±1.05	3
Mg	9.11	±11.06	2
Cu	2.5	±0.03	2
F	5.77	-	1
Cl	0.28	-	1
Fe	0.00	-	0
P	0.00	-	0
V	0.00	-	0

^1^ The number of implants contaminated with each element.

**Table 6 jfb-14-00172-t006:** Particle distribution on Zimmer Biomet dental implants. The shoulder had the higher particle count in most groups.

Zimmer Dental Implant														
Magnification	30×		100×				200× (Outer Surface)				200× (Inner Surface)			
Particle Size Area	L ^1^		M ^2^		S ^3^		LM ^4^		S		LM		S	
	Mean	SD	Mean	SD	Mean	SD	Mean	SD	Mean	SD	Mean	SD	Mean	SD
Shoulder	2.73	±0.89	1.80	±0.97	6.95	±3.8	1.55	±1.47	7.78	±2	0.34	±0.58	4.5	±3.74
Body	3.99	±1.48	1.37	±0.69	3.85	±2.27	0.99	±1.1	4.19	±2	0.20	±0.5	1.22	±1.02
Apex	4.01	±1.67	1.38	±1.31	5.97	±4	2.12	±1.5	3.97	±2.81	0.27	±0.55	3.13	±2.52
Implant mean	10.73	±2.2	4.56	±0.74	16.77	±4.74	4.66	±1.68	21.65	±1.5	0.81	±0.2	8.85	±4.95
*p*-value	<0.01		0.41		0.043		0.072		0.594		0.809		<0.01	

^1^ Large particles > 20 µm; ^2^ Medium particles Between 8 and 20 µm; ^3^ Small particles < 8 µm; ^4^ Large and medium particles > 8 µm.

**Table 7 jfb-14-00172-t007:** Elements distribution on implants and their concentration by mass (wt%) in the particles found on Cortex dental implants with EDS. Aluminum (Al) was found on three different implants, the shoulder was clean. Titanium (Ti) was found on two implants at the shoulder. Silica (Si) was detected on five different implants at seven different sites. Sulfur (S) was detected on nine different implants at ten different sites. Calcium (Ca) was detected on eight different implants. Sodium (Na) was detected on six different implants. Magnesium (Mg) was detected on two different implants at two different sites. Chlorine (Cl) was detected on one implant at the apex. Iron (Fe) was detected on one implant at the apex. Phosphate (P) was detected on one implant at the shoulder.

Zimmer			
**Elements**	Mean%	SD	(n) ^1^
C	37.52	±12.18	15
N	49	±13.76	15
Al	35.62	±42.85	3
O	21.15	±10.15	14
Ti	86.1	±3.13	2
Si	7.18	±11.5	5
S	0.64	±0.2	9
Ca	10.09	±5.37	8
Na	0.68	±0.27	6
Mg	8.89	±9.5	2
Cu	0.00	-	0
F	0.00	-	0
Cl	12.16	-	1
Fe	5.31	-	1
P	2.1	-	1
V	0.00	-	0

^1^ The number of implants contaminated with each element.

**Table 8 jfb-14-00172-t008:** The difference in particle number areal mean between implant brands. Zimmer Biomet dental implants and the S group had the highest particle numbers.

Particle Implant Size Brand	L ^1^		M ^2^		S ^3^	
	Mean%	SD	Mean%	SD	Mean%	SD
Cortex	1.46	±0.33	0.97	±0.21	1.89	±0.8
Keystone	1.80	±0.38	1.12	±0.3	3.33	±0.5
Zimmer Biomet	3.58	±0.73	1.52	±0.25	5.59	±1.6
*p*-value	<0.01		0.052		<0.01	

^1^ Large particles > 20 µm; ^2^ Medium particles Between 8 and 20 µm; ^3^ Small particles < 8 µm.

**Table 9 jfb-14-00172-t009:** Particle distribution Summary. The difference in the same area between implant brands. Cortex had the lowest particle numbers in most areas and at most magnifications.

Magnification	Area	Dental Implants Brand						
		Cortex		Keystone		Zimmer		*p*-Value
		Mean%	SD	Mean%	SD	Mean%	SD	
30×	Sh ^5^	1.63	±1.05	1.64	±1.7	2.73	±0.89	0.01>
	B ^6^	1.66	±1.35	2.24	±1.6	3.99	±1.48	0.01>
	A ^7^	1.08	±0.94	1.53	±0.88	4	±1.67	0.01>
100× M	Sh	1.17	±0.91	1.38	±1.08	1.80	±0.97	0.22
	B	1.01	±0.69	1.18	±1.1	1.37	±0.69	0.51
	A	0.74	±0.78	0.79	±0.83	1.38	±1.3	0.165
100× S	Sh	2.39	±2.28	3.72	±2.32	6.95	±3.8	0.01>
	B	2.32	±2.17	3.41	±2	3.85	±2.27	0.15
	A	0.97	±1.65	2.56	±2.6	5.9	±4.04	0.01>
200× Out S ^1^	Sh	3.3	±2.85	3.3	±2.1	7.78	±2.67	0.01>
	B	2.50	±2.2	4.19	±2	7.01	±2.5	0.01>
	A	1.9	±2.5	3.97	±2.8	6.86	±3.3	0.01>
200× Out ML ^2^	Sh	0.97	±1.2	0.8	±0.8	1.75	±1.6	0.097
	B	0.88	±0.9	1	±1.33	0.92	±1.1	0.93
	A	0.16	±0.43	0.89	±0.93	2.11	±1.45	0.01>
200× In S ^3^	Sh	2.9	±3.5	0.7	±1.3	4.51	±3.7	0.01>
	B	1.5	±2	1.1	±1.7	1.2	±1	0.80
	A	1	±2.5	0.76	±1.1	3.1	±2.5	0.01>
200× In ML ^4^	Sh	0.07	±0.26	0.13	±0.52	0.66	±1.36	0.13
	B	0.23	±0.48	0	-	0.2	±0.52	0.26
	A	0.07	±0.26	0.07	±0.26	0.32	±0.56	0.13

^1^ Particles smaller than 8 µm on outer surface with 200× magnification. ^2^ Particles larger than 8 µm on outer surface with 200× magnification. ^3^ Particles smaller than 8 µm on inner surface with 200× magnification. ^4^ Particles larger than 8 µm on inner surface with 200× magnification. ^5^ Shoulder of the implant. ^6^ Body of the implant. ^7^ Apex of the implant.

**Table 10 jfb-14-00172-t010:** The change in large particle numbers after exposing the implant to air for 10 min. There was no significant change in any implant regardless of the manufacturer.

Implant Company	Cortex		Keystone		Zimmer	
	(n ^1^ = 15)		(n = 15)		(n = 15)	
	Mean	SD	Mean	SD	Mean	SD
Before	2.73	±1.703	3.363	±1.97	6.37	±2.12
After	2.74	±1.715	3.376	±2	6.39	±2.14
*t*-test	*t* (14) = −1		*t* (14) = −1		*t* (14) = −1.72	
*p*-value	0.334		0.334		0.1	

^1^ The number of implants examined.

## Data Availability

The data presented in this study are available upon request from the corresponding author. The data are not publicly available due to privacy.

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
