# Peer review of "Assessing the Cleanliness of Dental Implants Using Scanning Electron Microscopy and Energy-Dispersive X-ray Spectroscopy Analysis—A SEM and EDS In Vitro Study"

_jfb, 2023, doi:10.3390/jfb14030172_

Round 1

Reviewer 1 Report

As you wrote, the cleanliness of implants is important for implant biointegration and you have used scanning electron microscopy and EDS to detect and determine the nature of particules found on the surface of implants. You confirm the presence of particles on the implant surface previously demonstrated by other authors, however, by comparing implants of 3 different brands. Do you have any conflicts of interest with these brands of implants? I didn’t see that in the article. I have added comments and suggestions in the pdf of your article. Do you know the preparation technique for the different implants? Could these particles simply come from manufacturing defects in the implants? Do you kow the concentration of these elements necessary to limit the biointegration of the implants? Zimmer implants have a higher particles rate. Is the biointegration of these implants lower than for other implants? How to avoid or limit the presence of these particles? Finally, do you have any suggestions ? I can only invite you to response to my suggestions and questions.

Yours sincerely

Reviewer 2 Report

The authors presented Assessing the cleanliness of dental implants using scanning electron microscope and Energy-dispersive X-ray spectroscopy analysis 

Few modifications and revisions are required before giving them a final recommendation.

Authors should contain the type of research in the title - although it refers to dental implants on the market, it is a pure in vitro study.

Please pay attention to the reference numbers and correct them - some are written in italics some without italics.

In the introduction section, the authors describe the factory method of cleaning Keystone implants - I suggest (if such information is available) writing what it looks like in the case of other described implants.

In the introduction section (line 86-90) authors wrote e.g.: "It is a common method for chemical surface analysis in research" - please include in this sentence that it is a method (EDX) very often used in dental research of both material samples inorganic by design (as in the case of this study) and organic (as in these studies: https://doi.org/10.1016/j.actbio.2020.04.008; https://doi.org/10.3390/polym12112655 ; https://doi.org/10.3390/polym13152418).

Please pay more attention to the use of the abbreviations EDS and EDX - one of them refers to the device, the other to the method, although it is often used interchangeably even in publications - I leave the introduction of this change or omission at the discretion of the authors.

Please pay attention to starting sentences with capital letters in the discussion section - in some places you get the impression that the sentences were corrected and not checked. 

The authors in lines 473-475 wrote: "There was no significant difference in large contaminant counts after 10 minutes of exposure. This may indicate that a particle-free, clean room environment is not obligatory for similar studies" - this is a far-reaching sentence. The authors of the current study would only be able to conclude this if they compared the results of the same study, but split, where half of the tested samples would be performed in a clean room and the other as in the protocol adopted here. Please correct this sentence so as not to mislead the reader that it results from this study or remove it. 

To sum up, this is a very interesting study, which is certainly suitable for publication after taking into account the above comments, the only doubt that arises after reading the work is the lack of use of a cleanroom to conduct the study - which I encourage the authors to do in future research.

Reviewer 3 Report

Dear Authors,

Please find below some observations and recommendations concerning your article entitled” Assessing the cleanliness of dental implants using scanning electron microscope and Energy-dispersive X-ray spectroscopy analysis”.

Title and Authors

Please follow the MDPI authors' guidelines.

Please provide a correspondent author.

In the Abstract section:

- Please follow the MDPI authors' guidelines concerning the abstract structure (no more than 200 words should be included, without headings).

- Please rewrite the abstract section

In the Introduction section:

- Please indicate the reference number in square brackets before dot.

- The introduction is more like a literature review type. It should be rewritten. Some information can be simplified.

- Please write the aim of the study and the hypotheses to the Introduction section but not as subchapters (remove ”1.1. Objectives and Specific Aims” and ”1.2. Hypothesis”)

In the Materials and Methods section:

2.7. Statistical analysis

- Please add the software (product, version, Manufacturer, City and State) used for statistical analyses.

- Please provide the p-value level of significance used for statistical analysis.

In the Results section:

- Table 1- please correct the last row:”P. V with p-value (same for tables 3, 5, 7-10)

- Table 5- please restructure it to conform to tables 1 and 3.

- Please provide the Energy-Dispersive X-ray Spectroscopy (EDS) graphs

- Please add a new subchapter” 3.5. Differences in particle counts after implants were exposed to room air”, or reorder the Results section in two subchapters like:

3.1. Testing the first hypothesis

3.2. Testing the second hypothesis

In the Discussion section:

- Please add the reference immediately after the author’s name

Eg. -line 374-” Rizo-Gorrita et al. [39] used 20 kV and areal sampling with 3000x magnification.”

In the Conclusion section:

- line 499- Try to avoid generalisation” all implants are contaminated”- better” Studied implants”

In the References section: please follow the styles recommended for MDPI journals.

Reviewer 4 Report

Dear authors, thank you for all the efforts conducted on the elaboration of the present  manuscript. I would like to present a few concerns and comments:

I recommend the authors to place the study design in the title. 

According to the journal guidelines, the abstract does not require sub-headings. 

I recommend the authors to place the keywords by alphabetic order. 

The in text citations are not according to the journal guidelines. 

The Introductions looks sound. 

How was the n=15 defined?

May the authors provide the SEM settings?

What was the area size screened in the EDS?

Which softwares were used to analise the SEM images and extract the EDS results?

Did the authors tested the normality of the results?

I recommend the authors to debate the advantages of using SEM and EDS in a Multimethod assessment such this one. I suggest the study DOI: 10.3390/ma15155288 to support it. 

May the authors debate the strength of the study?

May the authors give further studies perspectives?

May the authors debate the possible generalization of the results?

The final reference list is not according to the journal guidelines. 

Round 2

Reviewer 4 Report

Dear author, I have no more concerns.